# Structural analysis of mtEXO mitochondrial RNA degradosome reveals tight coupling of nuclease and helicase components

Michal Razew[1], Zbigniew Warkocki[2], Michal Taube[3], Adam Kolondra[4], Mariusz Czarnocki-Cieciura[1], Elzbieta Nowak[1], Karolina Labedzka-Dmoch[4], Aleksandra Kawinska[4], Jakub Piatkowski[4], Pawel Golik[2,4], Maciej Kozak[3], Andrzej Dziembowski [2,4] & Marcin Nowotny[1]

Nuclease and helicase activities play pivotal roles in various aspects of RNA processing and degradation. These two activities are often present in multi-subunit complexes from nucleic acid metabolism. In the mitochondrial exoribonuclease complex (mtEXO) both enzymatic activities are tightly coupled making it an excellent minimal system to study helicase–exoribonuclease coordination. mtEXO is composed of Dss1 3′-to-5′ exoribonuclease and Suv3 helicase. It is the master regulator of mitochondrial gene expression in yeast. Here, we present the structure of mtEXO and a description of its mechanism of action. The crystal structure of Dss1 reveals domains that are responsible for interactions with Suv3. Importantly, these interactions are compatible with the conformational changes of Suv3 domains during the helicase cycle. We demonstrate that mtEXO is an intimate complex which forms an RNA-binding channel spanning its entire structure, with Suv3 helicase feeding the 3′ end of the RNA toward the active site of Dss1.

[1] Laboratory of Protein Structure International Institute of Molecular and Cell Biology, Trojdena 4, 02-109 Warsaw, Poland. [2] Laboratory of RNA Biology and Functional Genomics, Institute of Biochemistry and Biophysics, Polish Academy of Sciences, Pawinskiego 5a, 02-106 Warsaw, Poland. [3] Faculty of Physics, Adam Mickiewicz University, ul. Umultowska 89, 61-614 Poznan, Poland. [4] Institute of Genetics and Biotechnology, Faculty of Biology, University of Warsaw, Pawinskiego 5a, 02-106 Warsaw, Poland. Correspondence and requests for materials should be addressed to M.N. (email: mnowotny@iimcb.gov.pl)

RNA degradation pathways play crucial roles in processing of various types of RNA, regulation of gene expression, and efficient removal of defective RNAs. The main executors of the RNA degradation in eukaryotes are processive exoribonucleases that often form macromolecular assemblies and can act from either end of RNA molecules[1]. Exoribonucleases that exhibit 3′-to-5′ directionality usually cooperate with RNA helicases that are believed to facilitate substrate recruitment and help unwind RNA. The mechanism of cooperation, however, is poorly understood because no structures of a productive RNA helicase–exoribonuclease complex have yet been solved.

A minimal system that is responsible for 3′-to-5′ decay is the yeast mitochondrial degradosome (mtEXO) complex[2]. The mtEXO complex is composed of two subunits: the nucleotide triphosphate (NTP)-dependent RNA helicase Suv3 and the 3′-to-5′ exoribonuclease Dss1[3]. The activities of both components are essential for the functioning of the complex in vivo, and mutations in either SUV3 or DSS1 lead to severe pleiotropic dysfunction of the mitochondrial gene expression system, with over-accumulation of excised intronic sequences, high-molecular weight precursors, and the depletion of mature transcripts[3,4]. Concomitant translation defects lead to irreversible loss of the mitochondrial genome in mtEXO-deficient strains[5]. The mtEXO complex can be reconstituted in vitro with a 1:1 Suv3: Dss1 stoichiometry. Biochemical studies revealed a remarkable functional interdependence of the nuclease and the helicase activities within mtEXO[3,6]. The 3′-to-5′ directional helicase activity of the yeast Suv3 protein, requiring a substrate with a 3′ single-stranded overhang, is detectable only in complex with Dss1. Dss1 individually has low RNase activity, and the robust nucleolytic activity of the entire degradosome complex nearly completely depends on ATP, an unprecedented feature that is not found in other known RNA-degrading systems. The exonuclease activity of the mtEXO complex is also affected by mutations that abolish the ATPase activity of Suv3[3].

The crystal structure of human SUV3 revealed many similarities with superfamily 2 (SF2) DExH helicases. However, because of its special features, SUV3 was classified as a separate branch[7]. The core of Hs-SUV3 forms a ring-like structure that consists of two RecA domains and a C-terminal all-helical domain. The additional N-terminal domain of an unknown function protrudes from the helicase core. RecA domains are a common element of SF2 helicases, but the N-terminal and C-terminal domains of Suv3 are dissimilar to the auxiliary domains of any other helicases. In addition, at the level of the protein sequence small differences in ATP- and RNA-binding motifs of Suv3 helicases set them apart from other SF2 enzymes[7].

Dss1 belongs to the RNR family of metal-dependent exoribonucleases that degrade RNA from the 3′ end, producing 5′ monophosphates[8] and leave a final 4–6 nt digestion product[9]. The crystal structures of several family members (bacterial RNase II and eukaryotic RRP44/DIS3 and DIS3L2) revealed a α/β fold of the catalytic RNB domain[10–12]. This domain forms a barrel with a channel inside that leads to the exoribonuclease active site. The top of the RNB domain and the entry to the channel is decorated by two N-terminal cold-shock domains (CSDs) that adopt a nucleic acid-binding OB fold with a five-stranded β-barrel[13]. An S1 RNA-binding domain with a similar OB fold[14] is also present in the C-terminus. Single-stranded RNA occupies the nucleic acid-binding channel inside the RNB domain. The active site is located at the opposite end of the channel relative to its entrance and interacts with the 3′-end of the substrate[10]. Both CSDs and the S1 domain contribute to RNA binding. Notably, Dss1 is a special member of the RNR family because of the tight functional interdependence with its helicase partner Suv3.

To understand the structure and mechanism of the yeast mitochondrial RNA degradosome, we present crystal structures of Dss1 nuclease and the mtEXO complex of the yeast Candida glabrata (Cg)[15]. We further complement these structural data with biochemical and biophysical experiments. Our results reveal structural features of Dss1 that mediate a stable interaction with Suv3. Moreover, based on our structural analyses and biochemical experiments, we explain the ways in which RNA is actively fed by Suv3 into the Dss1 RNB channel toward the nuclease catalytic center.

## Results

**Crystal structure of Cg-Dss1 reveals its domain architecture.** To gain insights into the structure and mechanism of the mtEXO complex, we first solved the crystal structure of Dss1 protein for which no structural information was previously available. Full-length Cg-Dss1 protein was unstable and we could not obtain its crystals. Thus, we performed secondary structure predictions and limited proteolysis experiments, which indicated that the N-terminal region of Dss1 was partially unstructured. We then designed several deletion constructs and obtained stable preparations of the Cg-Dss1 protein with a deletion of 69 N-terminal residues (Cg-Dss1$^{70–900}$) (Fig. 1a). We also introduced an inactivating point substitution (D477N) in the active site of the RNB domain to prevent RNA degradation during crystallization. We obtained crystals of this variant of Cg-Dss1 (Cg-Dss1$^{70–900}$ D477N) that belonged to space group P 1 and diffracted X-rays to 2.7 Å resolution at a synchrotron source. The structure was solved using crystals of selenomethionine-substituted protein and single-wavelength anomalous diffraction (SAD; Table 1). Two copies of the protein molecule were present in the asymmetric unit. The structure was refined to an $R_{free}$ of 24.0%, with excellent geometry of the model. The electron density maps are shown in Supplementary Fig. 1a, b.

The structure of Cg-Dss1$^{70–900}$ D477N is shown in Fig. 1b. It comprises a typical catalytic RNB domain. Compared with other RNase II-like enzymes, however, it has a unique composition of auxiliary domains. At the N-terminus a β-barrel domain is present (residues 92–170 of Cg-Dss1) with a patch of positively charged residues facing the RNA-binding channel (Fig. 1c). It is composed of 5 anti-parallel β-strands similar to the CSD1 of the other RNB nucleases. However, structural alignment performed using DALI[16] indicated that it is more similar to the KOW (Kyprides, Ouzounis, Woese) domain[17] which is found in a variety of RNA-interacting proteins. Interestingly the KOW-like domains are present in the exosome-associated helicases Mtr4 and Ski2[18,19] where they mediate binding of ribosomal RNPs. The β-barrel domain in Cg-Dss1 is more similar to the Ski2 helicase, since it lacks the KOW sequence motif.

The region that corresponds to CSD2 of RNase II-like enzymes in Cg-Dss1 contains two previously unidentified domains. The first of these domains is located between residues 170 and 320. DALI[16] search revealed that it is a winged helix (WH) domain. WH domains represent a subclass of helix-turn-helix (HTH) motifs with α1–β1–α2–α3–β2–β3 topology[20]. The loop between the β2 and β3 strands is known as the wing element. Compared with typical WH domains, the Cg-Dss1 WH has a longer N-terminal helix α1 (residues 189–227) that comprises a kink at Pro214. It also contains an additional fourth helix (residues 260–271) between α2 and α3. WH domains are known to bind nucleic acids and mediate protein–protein interactions[20,21]. In the case of Cg-Dss1 most of the region of the WH domain that could be involved in nucleic acid binding is blocked by extended helix α1 and the β-barrel domain, but some charged residues in this region face the RNA-binding channel of Cg-Dss1 and thus may participate in nucleic acid interactions.

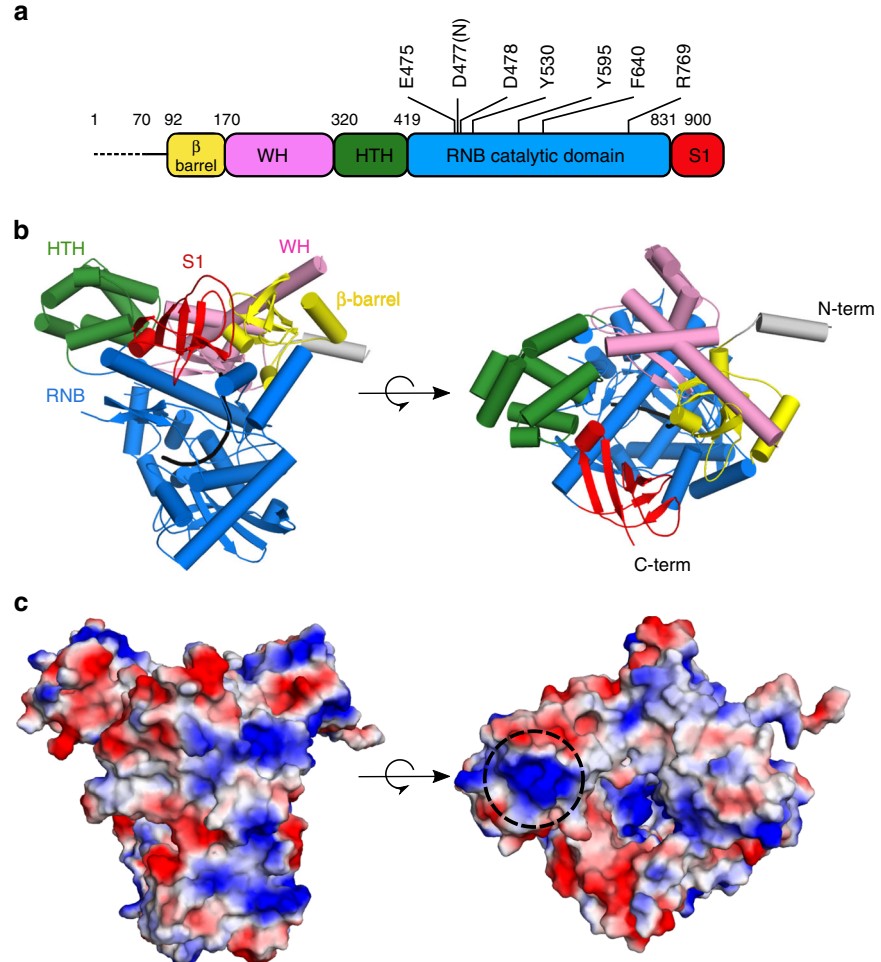

**Fig. 1** Crystal structure of *C. glabrata* Dss1$^{70-900}$ D477N. **a** Schematic of the domain composition of *Cg*-Dss1. The domains are shown in colors: β-barrel in yellow; WH (wing-helix) in pink; HTH (helix-turn-helix) in green; RNB (nuclease domain) in blue; S1 in red. The dashed line corresponds to the region deleted in the crystallized variant. The positions of the residues of the RNB domain that are involved in RNA binding and degradation are indicated. **b** Overall structure of *Cg*-Dss1$^{70-900}$ D477N. The domains are colored as in **a**. The co-crystallized RNA molecule is shown in black. **c** Electrostatic surface potential of *Cg*-Dss1$^{70-900}$ D477N generated in PyMol (positively charged regions in blue, negatively charged regions in red, ±65 kTe$^{-1}$). A positively charged patch located in the HTH domain is indicated by the dashed circle

The second domain that is unique to Dss1 is a double HTH domain (residues 320–419 in *Cg*-Dss1; Fig. 1a, b). In Dss1, this domain comprises a helical bundle (residues 360–413) that forms the canonical HTH that is preceded by three additional helices. The most common function of HTH domains is nucleic acid binding. A characteristic feature of HTH domains is an interaction between Arg/Lys residues and the major groove of the DNA[22]. Indeed, HTH domain of Dss1 comprises a positively charged patch on its surface (made by residues Lys387 and Arg390) that could aid in binding of nucleic acids (Fig. 1c). However, in the mtEXO complex, the residues forming this patch participate in protein–protein interactions (Fig. 2a–c) and are unlikely to interact with the nucleic acid. An HTH domain was also found in the N-terminus of the RNase II-type enzyme DrII from *Deinococcus radiodurans*, where it replaces both CSDs and forms an open RNA-binding surface that is capable of accommodating pre-tRNA (Supplementary Fig. 2e)[23]. In addition to the DrII HTH domain, the DALI search found another similar structure to the Dss1 HTH domain: a DNA-binding domain of the bacterial quorum-sensing transcription regulator SdiA[24] (PDB ID: 4Y13). In Dss1, the HTH and S1 domains interact with each other through an interface between a loop that connects the last two β-strands of the S1 and C-terminus of the first helix

of the HTH (Fig. 1b). The structure of *Cg*-Dss1$^{70-900}$ D477N comprises an additional unique element, an N-terminal helix (residues 77–90), that protrudes from the core of the protein (Fig. 1b).

The catalytic domain of *Cg*-Dss1 is very similar to the RNB domains of the other RNase II family members and can be superimposed on them with an RMSD between 1.6 and 2.0 Å (Supplementary Fig. 3a, b). *Cg*-Dss1 also possesses the S1 domain (residues 831–900 in *Cg*-Dss1; Fig. 1a, b) that has a structure and position very similar to S1 domains in other members of the family (Supplementary Fig. 2). Overall, among RNase II enzymes, the domains that decorate the RNB core are arranged in one plane, with the exception of the *Ec*-RNase II whose CSD1 is located further away from the catalytic domain, forming the so-called "anchoring region." In *Cg*-Dss1, the β-barrel domain that corresponds to the CSD1 has a unique placement attributable to the presence of the WH domain which in Dss1 occupies the position of CSD1 in Dis3L2 and Rrp44 (Supplementary Fig. 2).

In the structure we also observed an RNA molecule bound in the catalytic channel of *Cg*-Dss1 (described in more detail in Supplementary Note 1, Supplementary Fig. 4). This channel is overall positively charged and it leads to the active site. It is narrow and accessible only for ssRNA, which explains why Dss1

**Table 1 Data collection and refinement statistics**

| Data collection | SeMet-*Cg*-Dss1[70–900] D477N | *Cg*-Dss1[70–900] D477N Suv3[43–685] complex | *Cg*-Dss1[70–900] D477N SeMet-Suv3[43–685] | SeMet-*Cg*-Dss1[70–900] D477N Suv3[43–685] |
|---|---|---|---|---|
| Space group | *P* 1 | *C* 2 2 2₁ | *C* 2 2 2₁ | *C* 2 2 2₁ |
| Cell dimensions | | | | |
| *a*, *b*, *c* (Å) | 73.4, 82.2, 110.4 | 104.6, 151.2, 284.2 | 104.3, 152.3, 284.3 | 104.1, 152.5, 283.3 |
| *α*, *β*, *γ* (°) | 106.2, 106.6, 90.9 | 90.0, 90.0, 90.0 | 90.0, 90.0, 90.0 | 90.0, 90.0, 90.0 |
| Resolution (Å) | 29.2–2.7 (2.77–2.7) | 50.1–3.55 (3.64–3.55) | 50.1–3.79 | 50.1–3.97 |
| CC$_{1/2}$ | 99.4 (70.8) | 99.9 (97.6) | 99.9 (94.0) | 99.8 (90.2) |
| *I* / *σI* | 8.9 (1.5) | 27.4 (5.9) | 14.7 (3.9) | 9.5 (3.3) |
| Completeness (%) | 98.8 (88.8) | 95.4 (94.6) | 99.6 (100) | 95.5 (99.8) |
| Redundancy | 4.6 (2.8) | 4.3 (4.2) | 7.0 (7.1) | 6.7 (7.1) |
| Refinement | | | | |
| Resolution (Å) | 2.7 | 3.55 | | |
| No. reflections | 129386 | 26391 | | |
| $R_{work}$ / $R_{free}$ (%) | 19.6/24.0 | 29.6/32.8 | | |
| No. atoms | 12284 | 7574 | | |
| Protein | 11685 | 7455 | | |
| Nucleic acid/ion | 256/2 | 119 | | |
| Water | 144 | — | | |
| *B*-factors (Å$^2$) | 66.2 | 153.5 | | |
| Protein | 65.7 | 153.2 | | |
| Nucleic acid/ion | 69.0/114.3 | 169.2 | | |
| Water | 55.2 | — | | |
| RMSD | | | | |
| Bond lengths (Å) | 0.012 | 0.007 | | |
| Bond angles (°) | 1.291 | 1.440 | | |

is capable of digesting only single-stranded nucleic acids. The conformation of RNA and its mode of binding by the enzyme are very similar to other members of the RNR superfamily, indicating the same mechanism of substrate cleavage (Supplementary Fig. 4)[10]. Moreover, many of the residues involved in RNA binding are conserved among family members (Supplementary Fig. 3b) and some have been studied in detail for the RNase II enzyme[25].

In summary, the structure of Dss1 shows that it is a special member of the RNase II family in which N-terminal domains - β-barrel, WH and HTH replace the typically observed CSD1 and CSD2.

**Structure of mtEXO reveals the arrangement of the subunits**. To obtain structural and mechanistic details of the mtEXO complex, we performed crystallization trials with full-length complexes and several truncation variants of *Cg*-Dss1 and *Cg*-Suv3 (listed in Supplementary Table 1) in the presence and absence of RNA. The activity of these variants was tested in RNA cleavage assays (described in detail in the Supplementary Note 2, Supplementary Fig. 5). We obtained crystals for Dss1[70–900] D477N in complex with Suv3[43–685] in the presence of a dsRNA substrate with a 3′ 8 nt overhang (annealed 20Tx and 12Bx oligonucleotides; Supplementary Table 2) and ATP/Mg$^{2+}$. These crystals belonged to the *C* 2 2 2₁ space group (Table 1). The structure was solved by molecular replacement using two search models: (i) the refined *Cg*-Dss1 structure described above and (ii) a homology model of *Candida glabrata* Suv3 protein prepared with Swiss-Model[26] based on the published structure of the human enzyme[7] (PDB ID: 3RC3). The final model was refined at 3.55 Å resolution (Fig. 2a, b; Supplementary Fig. 1c). The tracing of the model was confirmed by calculating anomalous difference maps from the data sets that were collected for crystals of the mtEXO complex in which either Dss1[70–900] or Suv3[43–685] was selenomethionine-substituted (Supplementary Fig. 1d). We refer to this structure *Cg*-mtEXO for simplicity and use "full-length *Cg*-mtEXO" description, where applicable.

The structures of *Cg*-Dss1 alone and in the *Cg*-mtEXO complex are very similar (RMSD of 0.77 Å over 533 C-α atoms); therefore, Dss1 does not change its conformation upon interactions with Suv3. The structure of the *Cg*-Suv3 core is also very similar to *Hs*-SUV3, which was used to build the homology model for molecular replacement. *C. glabrata* and human helicases can be superimposed with an RMSD of 1.36 Å over 333 C-α atoms.

In the *Cg*-mtEXO structure, Suv3 is positioned close to the small accessory domains of Dss1: β-barrel, HTH, WH, and S1 which decorate the catalytic RNB domain and form a funnel around the entry to the RNA-binding channel. Two main interfaces are formed between *Cg*-Suv3 and the small domains (Fig. 2b, c). The first and much more extensive interface involves the HTH domain of *Cg*-Dss1 (residues Tyr338, Thr383, Ser386, Arg390, Asp397, and Val400; Supplementary Fig. 6a) and residues from the B-α-1′ helix of the first RecA domain of *Cg*-Suv3 and its vicinity (Asn187, Glu190, Lys197, and Arg199; Fig. 2c). The B-α-1′ helix is highly conserved among Suv3 proteins and not present in the canonical RecA domains in the SF2 superfamily of helicases[7]. Its participation in Dss1 interactions would explain its function. The involvement of the Dss1 HTH domain in this interface also explains the role of this domain that is unique for Dss1 protein.

To verify the importance of this interface, we performed mutagenesis studies, which are described in more detail in the Supplementary Note 3. Briefly, substitution of Ser386 or Arg390 of *Cg*-Dss1 to tryptophan did not disrupt mtEXO complex formation but altered its structure (Supplementary Fig. 7) and reduced its activity (Supplementary Fig. 6b–d). Moreover, in yeast *Saccharomyces cerevisiae* introduction of a Dss1 variant with substitution corresponding to S386W led to a strict respiratory deficiency (Supplementary Fig. 6e).

The second smaller Suv3-Dss1 interface forms between the C-terminal domain of the *Cg*-Suv3 helicase and a long kinked α-helix of the WH domain of *Cg*-Dss1. This interaction involves hydrophobic patches that comprise residues Trp210 and Leu213 of *Cg*-Dss1 and Met523 and Phe574 of *Cg*-Suv3. Additional Suv3-

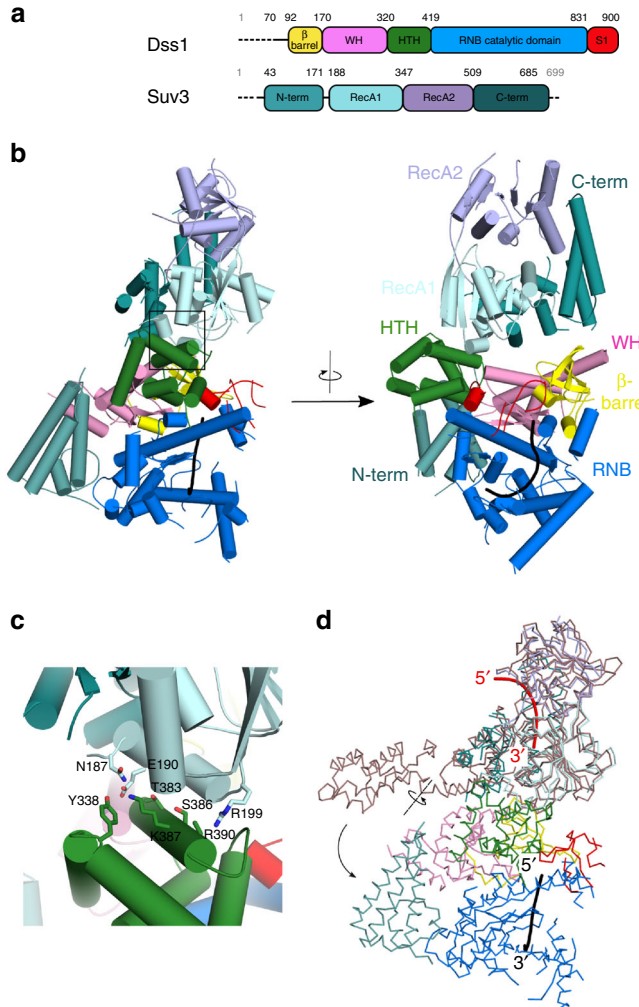

**Fig. 2** Structure of the *Candida glabrata* mtEXO complex. **a** Schematic of the domain composition of *Cg*-Dss1 and *Cg*-Suv3. The dashed line corresponds to the regions deleted in the crystallized variants. **b** Overall structure of *Cg*-mtEXO complex, comprising *Cg*-Dss1$^{70-900}$ D477N and *Cg*-Suv3$^{43-685}$. The domains are colored as in **a**. The co-crystallized RNA that is trapped in the RNB domain is shown in black. **c** Close-up view of the protein–protein interface that forms between the HTH domain of *Cg*-Dss1 and B-α-1′ helix of the RecA1 domain of *Cg*-Suv3, boxed in **b**. **d** Superposition of *Cg*-mtEXO and human SUV3 structure containing a 6 nt RNA chain (PDB ID: 3RC8), shown as a pink ribbon. The difference in the position of the N-terminal domain between *Cg*-Suv3 and *Hs*-SUV3 is shown with arrows. RNA chains in *Hs*-SUV3 and *Cg*-Dss1 are shown in red and black, respectively

Dss1 contacts in the crystal structure are mediated by the N-terminal domain (ND) of *Cg*-Suv3. Compared with human SUV3, this domain of *Cg*-Suv3 in the mtEXO complex has a different position relative to the rest of the structure. It is rotated by ~90° (Fig. 2d) around a hinge between residues 172 and 185 and interacts with a cleft between the WH and HTH domains and RNB domain of *Cg*-Dss1 (Fig. 2b). Few, mainly hydrophobic, contacts are formed between the ND of *Cg*-Suv3 and the RNB domain of *Cg*-Dss1. These interactions are mediated by Val439 and Leu443 of *Cg*-Dss1 and Met109 and Ile112 of *Cg*-Suv3. The sequence of the linker that connects the ND with the helicase core is conserved among higher eukaryotes and forms a single α-helix. However, this region is not conserved in fungi, and we did not observe its electron density in the *Cg*-mtEXO structure, suggesting that it was disordered. The flexibility of this linker

would allow the ND of *Cg*-Suv3 to adopt a very different position compared with the human protein.

Importantly, the arrangement of mtEXO subunits observed in our structure is in agreement with the polarity of nucleic acid binding by the helicase and nuclease subunits. When *Hs*-SUV3 is superimposed on *Cg*-mtEXO, the RNA bound by the human enzyme runs from 5′ to 3′ end toward the catalytic channel of *Cg*-Dss1, where the RNA observed in *Cg*-mtEXO structure continues toward the bottom of the RNB barrel and active site that binds the 3′ end of the nucleic acid (Fig. 2d).

In summary, the *Cg*-mtEXO structure reveals the arrangement of the helicase and nuclease components within the complex.

**Small-angle X-ray scattering confirms mtEXO architecture.** To test whether full-length (fl) *Cg*-mtEXO in solution adopts the architecture that we observed in the crystal structure of the deletion variant that had lower catalytic activity (Supplementary Fig. 5), we performed small-angle X-ray scattering (SAXS) experiments (Fig. 3 and Supplementary Fig. 8). We used fl-*Cg*-mtEXO, and the X-ray scattering curves were collected at three protein concentrations. A model of fl-*Cg*-mtEXO that contained unstructured regions that were not present or visible in the crystal structures was prepared using Bilbo (Fig. 3a)[27]. This model showed excellent agreement with the SAXS data for the fl-*Cg*-mtEXO complex with $\chi$ of 0.93. Moreover, SAXS-based three-dimensional reconstruction corresponded well to the overall shape of the *Cg*-mtEXO structure (Fig. 3b, d). The reconstruction had no prominent features that corresponded to the ND of *Cg*-Suv3, implying that it was mobile. To verify this, we collected SAXS data for mtEXO that comprised a variant of Suv3 with a deletion of the ND: *Cg*-Suv3$^{183–699}$. Three-dimensional reconstructions that were calculated from these data showed excellent agreement with the overall shape of the fl-*Cg*-mtEXO model and were similar to the reconstruction for the full-length protein (Fig. 3c). Furthermore, the ND of *Cg*-Suv3 was not essential for activity of the complex—the truncated complex (*Cg*-Dss1-Suv3$^{183–699}$) retained its activity in vitro (Supplementary Fig. 6). In *S. cerevisiae*, the truncated allele SUV3$^{215–737}$ complemented the loss-of-function phenotype of the Δsuv3 knockout, with similar respiratory growth behavior as the wildtype strain (Supplementary Fig. 5f). Therefore, the ND domain of Suv3 is not essential for mtEXO function and may be important for other roles of Suv3.

We next used the SAXS data for the fl-*Cg*-Dss-Suv3$^{183–699}$ complex to independently verify the arrangement of the complex subunits. We applied FoxsDock[28] to dock *Cg*-Suv3$^{183–699}$ to fl-*Cg*-Dss1 and obtained multiple models with various placements of *Cg*-Dss1 and *Cg*-Suv3. The model that corresponded to the orientation of both components that was observed in the crystal structure had the best fit to the SAXS data with $\chi$ of 0.93 (Fig. 3e, f).

In conclusion, the SAXS data confirmed that the full-length *Cg*-mtEXO in solution adopts the same subunit arrangement as in the crystal structure and are consistent with the ND of Cg-Suv3 being mobile with respect to the core of the complex.

**Biochemical experiments validate the RNA-binding channel.** The structural data for mtEXO imply that the RNA-binding channel spans the entire length of the complex from entry to Suv3 to the bottom of the catalytic barrel of Dss1 where the active site is located. Based on our crystallographic model, we estimated that the length of RNA that spans the entire channel from entry of the *Cg*-Suv3 helicase ring to the active site of *Cg*-Dss1 is ~18 nt. A 6 nt RNA fragment is present in the *Cg*-mtEXO structure inside the RNA-binding channel of Dss1 and it is superimposable with the RNA observed in *Cg*-Dss1 forming the same contacts with the protein (Supplementary Fig. 4b, c and Supplementary Note 1). A

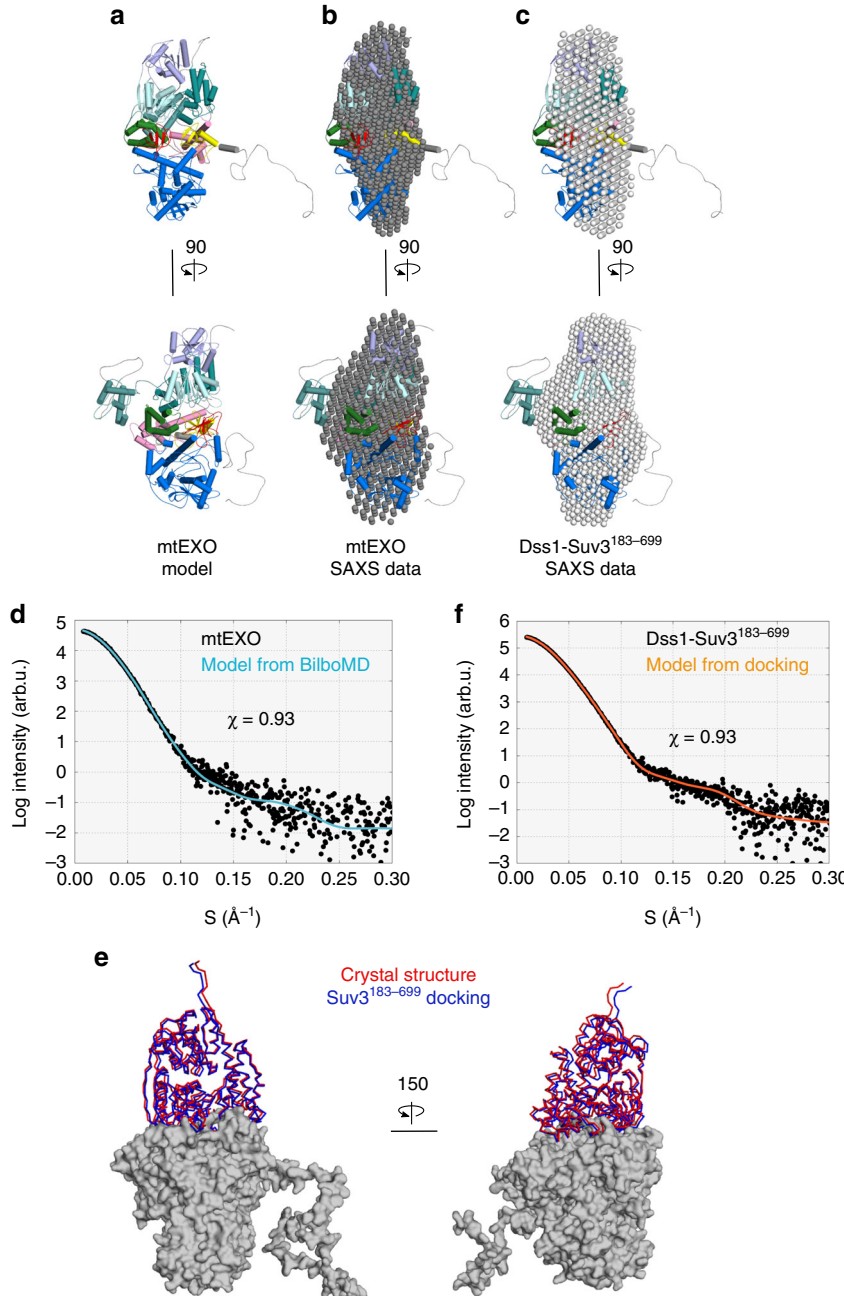

**Fig. 3** Small-angle X-ray scattering experiments for *Cg*-mtEXO. **a** Model of full-length (fl) *Cg*-mtEXO calculated using Bilbo. **b** Model of fl-*Cg*-mtEXO with superimposed three-dimensional reconstruction from SAXS data for fl-*Cg*-mtEXO (gray spheres). **c** Model of fl-*Cg*-mtEXO with superimposed three-dimensional reconstruction from SAXS data for *Cg*-Dss1-Suv3[183–699] (gray spheres). **d** Comparison of the computed X-ray scattering curve for fl-*Cg*-mtEXO model shown in **a** (cyan) with the scattering data collected for fl-*Cg*-mtEXO (black circles). **e** The best model obtained from docking *Cg*-Suv3[183–699] to *Cg*-Dss1 using FoxsDock[28]. *Cg*-Dss1 is shown in gray surface representation. The Suv3 subunit from the *Cg*-mtEXO crystal structure is shown as a red wire. The position of *Cg*-Suv3 in the best-scored model is shown as a blue wire. **f** Comparison of the computed scattering curve for the model from docking shown in **e** (orange) with the SAXS data for *Cg*-Dss1-Suv3[183–699] (black circles)

6 nt RNA fragment is also bound in the structure of human SUV3 with the first nucleotide exposed to the solvent[7]. The RNA is bound by residues from the helicase motifs: Ia (Lys234), Ib (Thr275) of the RecA1 domain and IV (Phe373 and Lys375), V (Thr424 and Asp425) of the RecA2 domain[7]. These motifs are conserved in *Cg*-Suv3 and we assume they will contact RNA in the same way. Upon superposition of *Cg*-mtEXO and *Hs*-SUV3 structures (Fig. 2d), the distance from the 3′ end of the SUV3-bound RNA to the 5′ end of the Dss1-bound RNA is 41 Å, corresponding to ~7 nt of fully stretched RNA. Therefore, the

protein-protected RNA length in the mtEXO channel would be ~18 nt.

To experimentally verify the length of the RNA that is bound inside the mtEXO complex, we performed exoribonuclease activity assays with a 5′-fluorescein-labeled ssRNA substrate in the presence of anti-fluorescein IgG antibody. We reasoned that binding of the antibody to the fluorescent label would stall RNA translocation within the catalytic channel, leading to the appearance of RNA fragments that result from aborted degradation. The length of these fragments would correspond to the

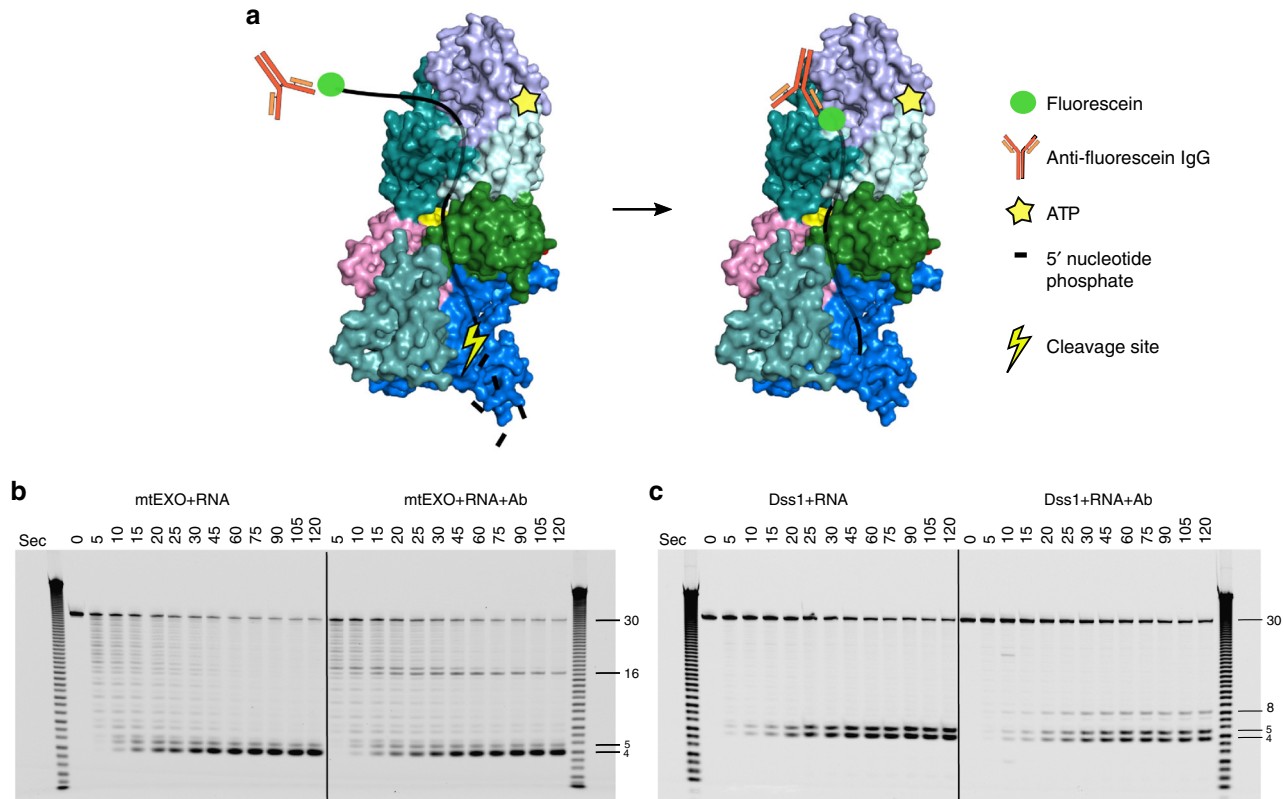

**Fig. 4** Biochemical validation of the length of RNA-binding channel within the *Cg*-mtEXO complex and *Cg*-Dss1 alone. **a** Schematic of the experiment with fluorescein-labeled RNA degradation by mtEXO in the presence of anti-fluorescein antibody that blocks RNA translocation. **b**, **c** Exoribonuclease activity assay of full-length *Cg*-mtEXO and *Cg*-Dss1, respectively, on the fluorescein-labeled substrate W30-F (see Supplementary Table 2 for sequence) in the presence and absence of the anti-fluorescein antibody. The reaction products at the time-points indicated on top of the gel were analyzed by 20% TBE-urea PAGE and scanned for fluorescent signal of the substrate

distance between the entry to the helicase ring and the nuclease active site (Fig. 4a). We performed time-course experiments of the degradation of a 5′-fluorescein-labeled ssRNA substrate by fl-*Cg*-mtEXO and fl-*Cg*-Dss1 with time-points between 5 s and 2 min in the presence or absence of an anti-fluorescein antibody. Consistent with our prediction, additional bands were observed in the reactions in which the antibody was present. For fl-*Cg*-mtEXO, in the case of both substrates, the length of this fragment was initially 17 nt, which after approximately 1 min was converted to a 16 nt fragment. We assume that the 17 to 16 nt truncation is a result of the cleavage of the last nucleotide located at the active site of the Dss1 nuclease (Fig. 4b). Therefore, the observed fragment length was 1 nt shorter than the length of RNA within the catalytic channel of *Cg*-mtEXO that was predicted based on the complex structure. For fl-*Cg*-Dss1 protein alone, the additional band observed in the presence of the antibody was 8 nt in length (Fig. 4c). This corresponds to the length of the RNA that can be accommodated in the RNA-binding channel of Dss1 that was predicted based on the RNase II and Dis3L2 structures in complex with longer RNAs.

**mtEXO but not Dss1 alone degrades structured RNAs**. The data presented so far support a model in which the RNA is directly fed by Suv3 ATP-dependent helicase activity into the Dss1 RNA-binding channel and toward the active site. To verify this, we used an RNase I footprinting experiment which revealed that protection of RNA fragments by the full-length-*Cg*-mtEXO occurs only in the presence of ATP, but not in its absence (Supplementary Fig. 9a, b and see Supplementary Note 4). In

contrast, the RNA footprint by Dss1 alone did not depend on ATP. This demonstrated the tight dependence of RNA binding by mtEXO on ATP hydrolysis by Suv3.

We next tested the role of helicase and nuclease coordination within mtEXO. We hypothesized that it is essential for the degradation of structured RNAs and general efficiency of RNA degradation. Furthermore, the rate of RNA degradation needs to be carefully controlled, and an ATP-dependent mechanism that relies on the availability of free nucleotides could be a possible solution. Previous work on mtEXO activity described its ability to digest dsRNA[4–6]. We verified that mtEXO from *C. glabrata* also cleaves dsRNA that contains a 3′ overhang only in the presence of ATP, whereas *Cg*-Dss1 that lacks intrinsic helicase activity is unable to process dsRNA (Supplementary Fig. 9c). However, no biochemical data are currently available on mtEXO activity toward structured RNA, such as excised group I introns, the accumulation of which is toxic to the cell[29]. Therefore, we investigated whether mtEXO has any further advantage over Dss1 alone in degrading structured RNAs. To test this, we combined 5′ $^{32}$P-labeled RNAs with either fl-*Cg*-mtEXO or fl-*Cg*-Dss1 alone in the presence of ATP/Mg$^{2+}$ and performed kinetic time-course exoribonuclease assays. We first analyzed RNA that does not form any secondary structure: R36 (see Supplementary Fig. 10 for sequence). Both fl-*Cg*-mtEXO and Dss1 degraded the RNA with comparable efficiencies and kinetics, with the substrate half-life of 0.76 min and 2.1 min, respectively (Fig. 5b; Supplementary Fig. 9d). We then tested the exoribonuclease activity of fl-*Cg*-mtEXO and fl-*Cg*-Dss1 on 5′ $^{32}$P-labeled RNAs that formed intramolecular base-pairs of variable lengths. We used 79-nt L1 RNA, L1 RNA with a 3′ 25-adenine tail, human vault RNA1–2

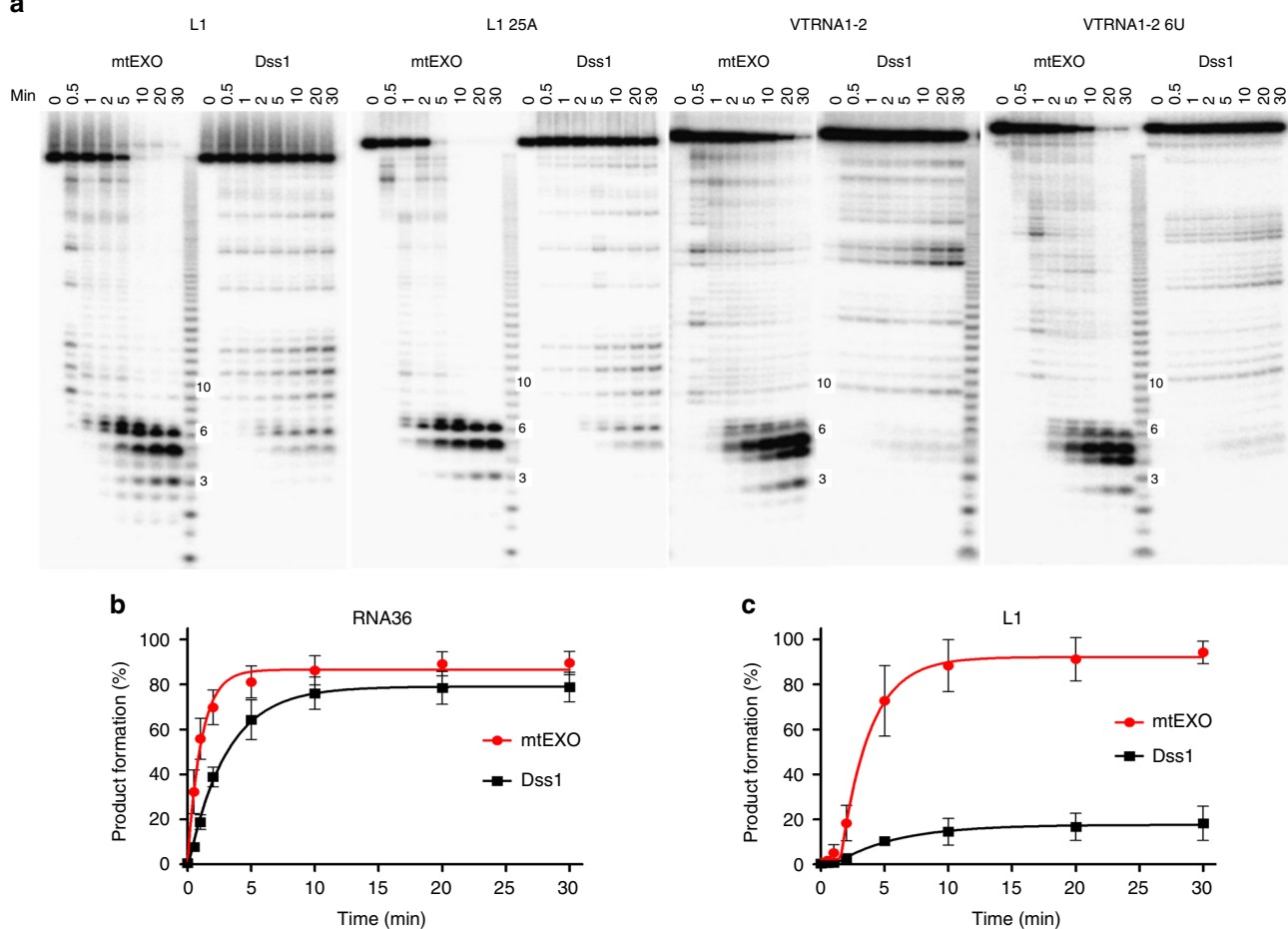

**Fig. 5** Degradation of complex RNA structures by *Cg*-mtEXO and *Cg*-Dss1. **a** RNA degradation assay with structured RNA substrates (listed in Supplementary Fig. 10) for full-length *Cg*-mtEXO or *Cg*-Dss1. Degradation products were analyzed by 18% denaturing TBE-urea PAGE. **b** Plot of R36 RNA degradation kinetics (see Supplementary Fig. 9d for images of PAGE analysis) by fl-*Cg*-mtEXO or fl-*Cg*-Dss1 in the presence of ATP (mean ± s.e.m. from three experiments). The final degradation products are shown as the sum of 2–5 nt RNA fragments. **c** Plot of L1 RNA degradation kinetics based on results shown in **a** by fl-*Cg*-mtEXO or fl-*Cg*-Dss1 in the presence of ATP (mean ± s.e.m. from three experiments). The final degradation products are shown as the sum of 2–6 nt RNA fragments

(89-nt), and human vault RNA1–2 with a 6-uridine tail (Fig. 5a, Supplementary Fig. 10). The degradation of these RNAs that formed intramolecular base-pairs was much more efficient with *Cg*-mtEXO than with *Cg*-Dss1 alone (Fig. 5c). In the case of *Cg*-Dss1, multiple intermediate stopping points were visible, demonstrating abortive degradation.

These results demonstrate the superior activity of *Cg*-mtEXO toward structured native-like and dsRNA substrates compared with *Cg*-Dss1 alone. This feature of *Cg*-mtEXO is consistent with the mechanism of helicase–nuclease coupling that was elucidated in our structural and biochemical studies and likely explains its role in discarding defective and excessive RNAs, in particular those with secondary structures.

## Discussion

We performed a complete structural and mechanistic characterization of the yeast mitochondrial RNA degradosome, a prototypic RNA helicase–exoribonuclease complex. Our data show that Dss1 is a unique family member of the RNase II family that comprises specialized WH and HTH domains that are responsible for interactions with Suv3 helicase. Our structural analysis revealed the architecture of the *Cg*-mtEXO complex, in which the helicase motor feeds the 3′ end of the RNA into the catalytic

channel of Dss1 for efficient and processive degradation. Substrate feeding by Suv3 would involve ATP binding, hydrolysis and release cycle, resulting in conformational changes of its RecA domains. These conformational changes have not been structurally described. However, structural information is available for various conformational states of another SF2 helicase, NS3[30]. Given the unusual features of Suv3 among SF2 enzymes, different modes of ATP/ADP binding between Suv3 and NS3, and different structures of the auxiliary domains of the two helicases, their comparisons are only an approximation. Nevertheless, the nature of the possible movements of the two RecA domains can be analyzed. Importantly, these analyses show that the movements of Suv3 can be accommodated with the mtEXO complex. One can envisage that RecA1 and C-terminal domains of Suv3 form a relatively rigidly placed platform that interacts with Dss1. RecA2, which does not form contacts with Dss1, is free to move during ATP hydrolysis and the resulting RNA translocation. Analyses of the AMPPNP-bound structure of *Hs*-SUV3[7] (PDB ID: 3RC8) indicate that the conformational state of Suv3 in our mtEXO complex structure corresponds to the ATP-bound state. We predict that upon nucleotide hydrolysis and release RecA2 domain of Suv3 would move away from RecA1 and Suv3-Dss1 interface and Dss1-Suv3 contacts will be maintained during RNA translocation (Fig. 6).

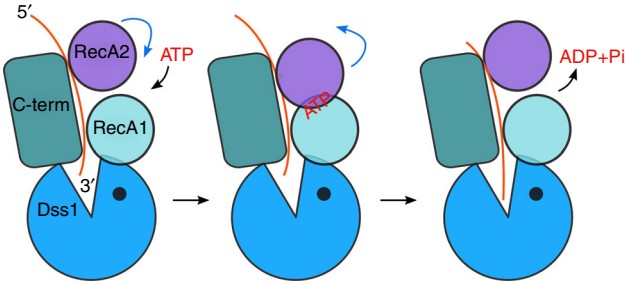

**Fig. 6** Proposed mechanism of RNA degradation by *Cg*-mtEXO. The cycle of ATP binding, hydrolysis and ADP release (black arrows) induces conformational changes of Suv3 helicase (blue arrows). The RecA2 domain (purple) would move toward the RecA1 domain (cyan) and away from it (blue arrows), while Suv3-Dss1 contacts would be maintained. The movements of RecA domains lead to the translocation of the 3′ end of the RNA (orange) into the RNA-binding channel and toward the active site of Dss1 nuclease (blue)

The mechanism described above is in line with the properties of Dss1 variants with point substitutions which do not disrupt the mtEXO complex but change its structure (Supplementary Note 3). The fact that these substitutions also inhibit the activity implies that the helicase and nuclease subunits must not only interact but also need to be properly aligned. This alignment is required so that Suv3 can precisely feed the RNA into the channel in Dss1. In addition, mtEXO complex needs to accommodate the conformational changes of the Suv3 helicase in the ATPase cycle (Fig. 6). If one of the two main contact points between helicase and nuclease are lost and the rigidity of the complex is reduced, the movements of Suv3 can lead to even more pronounced misalignment of the two subunits and result in defects in mtEXO function.

The concerted action of Suv3 and Dss1 is particularly important for structured RNAs which cannot be degraded by the nuclease on its own and for which helicase unwinding activity is required. In higher eukaryotes, including humans, the functional equivalent of the mtEXO degradosome is a complex of human SUV3 and PNPase[31,32]. Its architecture needs to be confirmed by detailed structural studies, and comparison of its mechanism to the one described herein for mtEXO will be informative.

The mtEXO complex is an interesting example of how helicase and nuclease activities can be combined. Joint action of these two activities is required in various systems, for example in RNA degradation[33–35], CRISPR[36], and DNA repair[37–39]. However, the mechanisms of helicase–nuclease cooperation vary greatly. mtEXO is unique because it shows exceptionally tight and intimate coupling of its nuclease and helicase at both the structure and activity levels.

An important issue is the way in which such a ubiquitous process as RNA degradation is regulated. RNA is generally targeted for degradation by specific markers that are recognized and processed by degradation machinery. In yeast, defective nuclear RNA is polyadenylated by the TRAMP complex and handed to the exosome for degradation. Other known markers are poly(U) tails on the 3′ end of miRNA that are specifically recognized and degraded by the Dis3L2 exoribonuclease[12,40,41]. In mammalian mitochondria, mRNA is also polyadenylated, and the degradation rate by the Suv3–PNPase complex depends on the length of the poly(A) tail, which correlates with the concentration of the inorganic phosphate in the mitochondrial matrix[31]. In yeast, mitochondrial RNA is not polyadenylated; therefore, a different mechanism of controlling the rate of RNA degradation needs to be present. Mature mRNAs end with a conserved dodecamer sequence that is recognized by dodecamer-binding protein that stabilizes the transcripts upon phosphorylation[42]. This

mechanism could also be coupled with ATP-dependent activity of the mtEXO complex, thus enabling precise control over RNA degradation.

In conclusion, we presented the first structural characterization of the mtEXO complex, a macromolecular machinery with very tight coupling of helicase and nuclease components. The entry to the RNA-binding channel of Dss1 is occupied by Suv3 which actively feeds the RNA into the channel for hydrolysis. This cooperation allows for the fast and efficient degradation of RNAs, particularly those that contain secondary and tertiary structures.

## Methods

**Protein expression and purification**. Expression in *E.coli* and purification of the full-length Dss1 proteins from several fungal species were tested, and the highest protein yields were obtained for *C. glabrata* and *S. cerevisiae*. Dss1 and Suv3 genes from *C. glabrata* and *S. cerevisiae* were cloned into the expression vector pDEST-His$_6$MBP with a TEV protease cleavage site between the His$_6$-MBP tag and the gene sequence (Supplementary Table 1). Mutations were introduced in the plasmids using QuikChange II Site-Directed Mutagenesis kit (Agilent). Full-length Dss1, Suv3, and their variants were expressed in the *E. coli* BL21(DE3)-RIL strain (Agilent). The cultures were grown at 37 °C in Luria broth (LB) medium, induced with 0.4 mM β-D-1-thiogalactopyranoside (IPTG) at OD$_{600}$ = 0.6–0.9, grown overnight at 18 °C, and harvested by centrifugation. Dry pellets were stored at −20 °C. Selenomethionine-substituted *Cg*-Dss1$^{70–900}$ D477N and *Cg*-Suv3$^{43–685}$ proteins were expressed in SelenoMethionine Expression Media (Molecular Dimensions) using the same protocol.

All of the Dss1 and mtEXO variants (both truncations and point substitutions) were purified using the same protocol. For mtEXO complex purification, pellets from bacteria that expressed the desired Dss1 and Suv3 variant were mixed and lysed by sonication in buffer that contained 50 mM Tris-HCl (pH 7.5), 0.5 M NaCl, 20 mM imidazole, 5% glycerol, and 5 mM 2-mercaptoethanol (buffer A). The lysate was clarified by centrifugation at 185,700 × *g* and the supernatant was loaded on a HisTrap column (GE Healthcare) that was equilibrated in buffer A. After washing with buffer A that contained 40 mM imidazole, the proteins were eluted with buffer A that contained 300 mM imidazole. His$_6$-MBP tag was removed by overnight incubation with TEV protease. Following ammonium sulfate precipitation and centrifugation at 17,400 × *g*, the pellet was dissolved in buffer A and reapplied on a HisTrap column. The flow-through that contained tag-less proteins was collected and applied to a Superdex 200 size-exclusion column (GE Healthcare) that was equilibrated with buffer B that contained 20 mM Tris-HCl (pH 7.5), 150 mM NaCl, 5% glycerol, and 1 mM dithiothreitol (DTT). The peak fractions that contained the desired protein/complex were concentrated on Amicon centrifugal filters and used for crystallization trials. For activity assays and long-term storage, proteins were additionally purified by gel filtration in buffer B that contained 0.5 M NaCl and 10% glycerol and kept at −80 °C.

**Crystallization**. Crystallization trials were performed at 18 °C using the sitting-drop vapor diffusion technique. Initial crystallization hits for the selenomethionine-labeled *Cg*-Dss1$^{70–900}$ D477N were identified in Crystal Screen (Hampton Research). After the optimization of crystal growth, the best diffracting crystals were obtained for 7 mg ml$^{-1}$ *Cg*-Dss1$^{70–900}$ D477N supplemented with 0.5 mM MgCl$_2$ and mixed with an equal volume of reservoir buffer that contained 0.1 M MES monohydrate (pH 6.5) and 6% (w/v) PEG 20000 using the hanging-drop vapor diffusion method. Crystals were cryoprotected in 30% glycerol and flash frozen in liquid nitrogen.

The mtEXO complex (*Cg*-Dss1$^{70–900}$ D477N and *Cg*-Suv3$^{43–685}$) was concentrated to 8 mg ml$^{-1}$. Prior to crystallization, it was mixed with the RNA substrate (annealed 20Tx: 5′-AUAAAAUAAAUAUUCUUAAUU-3′; and 12Bx: 5′-AUAUUAUUUUAU-3′) at a 1:1.1 molar ratio and supplemented with 0.5 mM MgCl$_2$ and 0.5 mM ATP. Annealing of the oligonucleotides resulted in a 12 bp duplex region with a 3′ 8 nt overhang. Sitting-drops were set up at 18 °C with an equal volume of the protein complex and reservoir solution. Optimized crystallization conditions from the Index Screen (Hampton Research) contained 0.2 M ammonium citrate tribasic (pH 7.0) and 18% (w/v) PEG 3350. Native and selenomethionine derivative crystals that were grown using the hanging-drop method were cryoprotected in 30% glycerol and flash frozen in liquid nitrogen.

**Structure solution and refinement**. X-ray diffraction data for selenomethionine *Cg*-Dss1$^{70–900}$ D477N crystals were collected at beamline I04 at Diamond Light Source, UK. The crystals belonged to space group *P* 1 and diffracted X-rays to ~2.7 Å. Because of radiation damage, data sets were collected from different regions of the crystal. Diffraction data were processed and scaled with XDS[43]. To improve the signal-to-noise ratio and multiplicity, three data sets that were collected at selenium peak wavelength (0.9785 Å) were merged in Aimless from the CCP4 Program Suite[44,45]. The *Cg*-Dss1$^{70–900}$ D477N structure was solved using SAD in the AutoSol module in Phenix[46] with two protein molecules in the asymmetric unit. Manual model building was performed in COOT[47] and the structure was refined in

phenix.refine[46] with 5% of unique reflections flagged for $R_{free}$ calculation. In the final model, 96% of the residues were in the most favored region of the Ramachandran plot, and 0.26% were localized in the disallowed region according to MolProbity[48]. For the purpose of refinement, the RNA chains were traced as 5′-CACUGA-3′ and 5′-AGAUAC-3′ for molecules I and II, respectively. However, because the RNA fragments co-purified with the protein, we expected them to have a mixture of various sequences.

Initial X-ray data sets for the mtEXO complex (Cg-Dss1[70–900] D477N and Cg-Suv3[43–685]) were collected at beamline 14.1 at Berliner Elektronenspeicherring-Gesellschaft für Synchrotronstrahlung in standard collection mode. A final dataset with helical crystal rotation was collected at microfocus beamline 23-2 at the European Synchrotron Radiation Facility, Grenoble. The structure was solved by molecular replacement in PHASER[49] using the previously solved structure of Cg-Dss1[70–900] D477N and the Cg-Suv3 homology model that was prepared in SwissModel[26] based on the human SUV3 structure (PDB ID: 3RC3) as search models. In the diffraction data, spots were visible up to 3.1 Å, but inspection of the electron density maps after the structure was solved indicated a lower resolution. Refinement was performed at different resolution cutoffs, and the 3.55 Å value was selected based on the fact that the inclusion of higher resolution data did not improve the appearance of the maps. The model was built in COOT[47] and refined using phenix.refine[46], including reference model restraints for low-resolution refinement. The Ramachandran plot of the final model included 92% residues in the most favored region and 1.2% residues in the disallowed region. Structural analyses of Cg-Dss1[70-900] D477N and Cg-mtEXO were performed in PyMol (PyMOL Molecular Graphics System, version 1.8, Schrödinger, LLC).

The Cg-mtEXO structure was solved at relatively low resolution. To confirm the tracing of the model, we produced crystals of the Dss1[70–900]–Suv3[43–685] complex, in which either Dss1 (14 methionines) or Suv3 (21 methionines) was substituted with selenomethionine (when both proteins were SeMet-modified, the crystals did not diffract X-rays). X-ray diffraction data were collected at selenium peak wavelength at Deutsches Elektronen-Synchrotron, Hamburg. Anomalous difference maps were calculated in Phenix to localize the selenium atoms of methionine residues. Anomalous difference peaks were observed for 30 of 35 methionines that were present in our model, confirming the correctness of our tracing.

The electron density maps for the N-terminal portion of Cg-Suv3 were of poor quality, and the low sequence conservation of this domain between Homo sapiens and C. glabrata made it very difficult to use the human structure to trace this part of the mtEXO complex. It contains five methionines, so anomalous difference maps could be used to guide model building for this region. However, because of the poor quality of the electron density maps, the ND structure of Cg-Suv3 has to be treated with caution. Although RNA was present in the crystallization drops, we did not observe electron density for the nucleic acid, with the exception of a 6 nt fragment near the active site of Cg-Dss1, which co-purified with Cg-Dss1 protein, similar to the structure of Cg-Dss1[70-900] D477N alone. The RNA that was added to the crystallizations of mtEXO apparently promoted the formation of crystals but was not incorporated in the structure.

**Small-angle light scattering data collection**. X-ray scattering experiments for the complexes of full-length Cg-mtEXO and a complex with a truncated ND of Suv3 (Cg-Dss1-Suv3[183–699]) were performed using the laboratory SAXS system Xeuss 2.0 (XENOCS, Grenoble, France), which was equipped with a MetalJet D2 microfocus X-ray generator (0.134 nm wavelength). Protein samples were injected in the low-noise liquid sample cell and incubated during measurements at 293 K. The details for the samples are shown in Supplementary Fig. 9. SAXS data were collected in 12–18 subsequent 10-min frames (total exposure time = 2–3 h) using a Pilatus3 1 M silicon pixel detector (Dectris, Switzerland). Data reduction and buffer subtraction were performed using the Foxtrot package. During the processing procedure, all of the frames were checked for radiation damage. Structural parameters that characterized the studied complexes (radius of gyration [$R_g$], Porod volume) were calculated in Primus from the ATSAS package[50]. The pair distance distribution function (P[R]) was calculated using Gnom[51].

**Molecular modeling of SAXS data**. Low-resolution models of Cg-mtEXO and Cg-Dss1-Suv3[183–699] complexes were obtained using the ab initio modeling program Dammin[52] based on the experimental SAXS data. For each studied sample, at least 10 independent models were generated and subsequently averaged and filtered using Damaver[53].

The full-length protein structures from the Cg-mtEXO complex for the molecular dynamics simulation were built in Modeller 9.17[54]. The Cg-Dss1[70–900] D477N crystal structure was used as a template, and missing parts were added in Modeller. The generated models were superimposed on the corresponding subunits from the crystal structure of the Cg-mtEXO complex. Residues 43–172 and 185–684 from Cg-Suv3 and 77–900 from Cg-Dss1 were excluded from the conformational sampling procedure. These parts cover rigid portions of the complex. To allow conformational sampling of the ND of the Cg-Suv3 protein, we defined the ND as a rigid body, connected by a flexible linker (residues 173–184) with the core of the complex. An alternative conformation of the Cg-Suv3 helicase core in the complex was also generated. The conformational search was performed using the BilboMD web server with 800 models per $R_g$ value[27].

**Macromolecular docking based on SAXS data**. The docking of Cg-Dss1 and Cg-Suv3[183–699] was performed using the FOXS Dock web server with default settings[28]. For docking, we used models of full-length Cg-Dss1 and N-terminally truncated Cg-Suv3[183–699] protein separated in space and SAXS data for the Cg-Dss1-Suv3[183–699] sample at 7 mg ml[−1].

**SEC-MALS analysis**. The molecular masses of Cg-mtEXO mutants of the Dss1 HTH-Suv3 RecA1 interface were determined by size-exclusion chromatography with multi-angle light scattering (SEC-MALS, Supplementary Fig. 7). Recombinant proteins and protein complexes (100 µl, 1 mg ml[−1]) were fractionated at room temperature on Superdex 200 Increase 10/300 column (GE Healthcare) equilibrated with SEC buffer that contained 20 mM Tris-HCl (pH 7.5), 150 mM NaCl and 1 mM DTT at 0.5 ml min[−1]. Elution of proteins was monitored by the following in-line detectors: UV 280/254 nm (1220 Infinity LC, Agilent Technologies), light scattering (DAWN HELEOS II, Wyatt Technology) and differential refractometer (Optilab T-rEX, Wyatt Technology). Data analysis and molecular weight calculations were performed using ASTRA 6 software (Wyatt Technology) using differential refractive index for a concentration calculation.

**RNA degradation assays with FAM-labeled RNAs**. RNA degradation assays were performed in buffer that contained 10 mM Tris-HCl (pH 7.5), 150 mM NaCl, 1 mM DTT, 1 mM MgCl₂, and 1 mM ATP at 30 °C with 60 nM protein and 30 nM RNA. The reaction was stopped at the selected time-points by the addition of loading buffer (95% formamide and 20 mM EDTA). Reaction products were resolved by 20% denaturing PAGE and visualized with Typhoon imager (GE Healthcare). Cg-Dss1 or Cg-mtEXO activity was quantified in ImageQuant TL 7.0 by dividing the amount of the final cleavage product by the total RNA signal from each lane. In the anti-fluorescein antibody protection experiment, the Monoclonal Anti-Fluorescein (FITC) IgG CF™ 488 A antibody (Sigma-Aldrich) was incubated with the 5′-fluorescein-labeled RNA substrate for 5 min at a final concentration of 2 µg ml[−1] prior to the degradation assay, which was conducted as described above. A list of substrates that were used for the activity tests and their sequences are presented in Supplementary Table 2.

**RNA degradation assays with ³²P-labeled RNAs**. Substrate RNA was either chemically synthesized (R36) or obtained by standard in vitro transcription using polymerase chain reaction-amplified templates[40] (Supplementary Fig. 10b). Substrate RNAs were 5′ ³²P-labeled using PNK (New England Biolabs, USA), followed by PAGE purification. Cold RNA with a trace of ³²P-labeled RNAs was dissolved to 300 nM in TNMgD buffer (50 mM Tris-HCl [pH 8.0], 150 mM NaCl, 3 mM MgCl₂, and 2 mM DTT) with or without 2 mM ATP as indicated. Full-length (fl) Cg-mtEXO or Cg-Dss1 was dissolved in TNMgD buffer to 350 nM for single-turnover conditions. For multiple-turnover experiments, fl-Cg-mtEXO, or fl-Cg-Dss1 was dissolved to 50 nM in TNMgD buffer. The RNA and protein mixtures were equilibrated to 30 °C and rapidly mixed in a 1:1 volumetric ratio (15 µl + 15 µl) to initiate the reactions. Aliquots of 2.5 µl were withdrawn at the indicated time-points and stopped rapidly by mixing with 5 µl ice-cold RNA loading dye (90% formamide, 20 mM Tris [pH 8.0], 10 mM EDTA, 0.1% sodium dodecyl sulfate (SDS), 0.025% [w/v] bromophenol blue, and 0.025% [w/v] xylencyanol). The samples were analyzed on 18% 19:1 denaturing gels (20 × 30 cm) in 1×TBE.

RNA mobility reference ladders were generated by alkaline hydrolysis of 5′ ³²P L1 or R36 RNAs at 98 °C in 50 mM sodium carbonate buffer in the presence of 1–3 µg yeast tRNA (Ambion) for 16, 20, or 27 min. Fragmentation was stopped by the addition of four volumes of RNA loading dye.

**RNase I RNA footprinting assays**. Body ³²P-labeled L1 RNA substrate (Supplementary Fig. 10) was generated by standard in vitro run-off transcription on a PCR-amplified template that contained a T7 consensus promoter and flanking duplex-stabilizing nucleotides. RNAs were PAGE-purified.

The RNase I protection assays were performed with 100 nM RNA substrates and 150 nM wildtype fl-Cg-mtEXO or fl-Cg-Dss1 in buffer that contained 20 mM Tris (pH 7.4), 50 mM NaCl, and 3 mM MgCl₂ with or without 2 mM ATP as indicated. RNA and proteins were incubated at 30 °C. Aliquots of 5 µl were withdrawn and mixed with 5 µl of TEN buffer (20 mM Tris [pH 7.4], 50 mM NaCl, and 5 mM EDTA). RNase I (Ambion, 1 µl, 100 U µl[−1]) was added and incubated for 18 min at 30 °C. The RNase I reaction was stopped by the addition of TNES (20 mM Tris [pH 7.4], 50 mM NaCl, 5 mM EDTA, and 1% SDS) supplemented with 0.1 µg µl[−1] proteinase K (USB) and 50 ng µl[−1] E. coli tRNA (Roche) and incubated for 30 min at 37 °C. RNAs were purified twice by phenol-chloroform extraction (100 µl polar phase volume) and ethanol precipitation. RNAs were recovered and separated in 18% 19:1 polyacrylamide gel (20 × 40 cm).

**Plasmid construction for complementation experiments**. Wildtype DSS1 ORF from S. cerevisiae was amplified using the primers DSS1_slicL and DSS1_slicR (Supplementary Table 3) and cloned into the centromeric shuttle vectors YCplac33 (URA3) and YCplac111 (LEU2)[55] using the one-step sequence- and ligation-independent cloning (SLIC) method[56] to yield YCplac33:DSS1 and YCplac11:DSS1 plasmids, respectively. Similarly, wildtype SUV3 ORF from S. cerevisiae was amplified using the primers SUV3_slicL and SUV3_slicR (Supplementary Table 3)

and cloned into YCplac33 (URA3) and YCplac111 (LEU2) using the SLIC method to yield YCplac33:SUV3 and YCplac111:SUV3 plasmids, respectively.

**Yeast strain construction.** The dss1::KanMX4 cassette was amplified from DNA of the EUROSCARF strain Y10873 (MATα; ura3Δ0; leu2Δ0; his3Δ1; lys2Δ0; YMR287C::kanMX4) with the primers DSS1_A and DSS1_D (Supplementary Table 3) and used to transform the wildtype diploid S. cerevisiae (W303 nuclear background, intronless mtDNA) strain YAK241 (MATa/MATα, ade2/ade2; trp1/trp1; ura3/ura3; leu2/leu2; his3/his3; [ρ+, intronless]), yielding the heterozygous deletant strain DDSSHZ (MATa/MATα; ade2/ade2; trp1/trp1; ura3/ura3; leu2/leu2; his3/his3; DSS1/dss1::kanMX4; [ρ+, intronless]). This strain was then transformed with YCplac33:DSS1. The transformant strain was then sporulated using a standard tetrad dissection protocol (Singer MSM200 dissection microscope). The haploid DSS1 knockout strain with the YCplac33:DSS1 maintenance vector was then selected on CSM-URA dropout medium (Formedium) with G418 (Sigma). The SUV3 deletion strain was constructed using the same procedure and the suv3::KanMX4 cassette that was amplified from DNA of the EUROSCARF strain Y12799 (MATα; ura3Δ0; leu2Δ0; his3Δ1; lys2Δ0; YPL029W::kanMX4) with the primers SUV3_A and SUV3_D (Supplementary Table 3).

**Yeast strains and growth conditions.** Δdss1 S. cerevisiae strains that expressed WT, A443W, S446W, and R450W substitution mutant alleles of Sc-Dss1 and Δsuv3 S. cerevisiae strains that expressed WT and the truncated allele of the Sc-Suv3 without the ND (Sc-Suv3$^{215-737}$) were grown in complete synthetic medium (CSM) without leucine. Overnight cultures were spotted in a series of 10-fold dilutions, starting at an optical density at 600 nm (OD$_{600}$) = 1 on YPD (glucose) and CSM-Leu with glycerol plates and incubated at 30°C for 3 days. Growth on CSM-Leu with glycerol plates is an indicator of respiratory competence. Representative results of six repeats that were obtained in two independent transformations are shown. Strains that were transformed with the empty vector were used as the negative controls (Δdss1 and Δsuv3).

**Site-directed mutagenesis for complementation plasmids.** Mutations in the S. cerevisiae DSS1 and SUV3 coding sequences were introduced into YCplac11:DSS1 using a modified Quikchange Site Directed Mutagenesis protocol (Agilent) and Phusion polymerase (Thermo Scientific) in the amplification step for 12 cycles (DSS1) or 16 cycles (SUV3), followed by DpnI (Thermo Scientific) digestion. The following mutagenic primer pairs were used at 125 ng per reaction: DSS1 A443W mutation (A443W_forward and A443W_reverse), DSS1 S446W mutation (S446W_forward and S446W_reverse), DSS1 R450W mutation (R450W_forward and R450W_reverse), and SUV3 32–214 deletion (MdSUV3_forward and MdSUV3_reverse). For oligonucleotide sequences see Supplementary Table 3.

All of the constructs were verified by Sanger sequencing that was performed in the Laboratory of DNA Sequencing and Oligonucleotide Synthesis, Institute of Biochemistry and Biophysics, Polish Academy of Science.

**Data availability.** Coordinates and structure factors for Cg-Dss1$^{70-900}$ D477N and Cg-mtEXO (Cg-Dss1$^{70-900}$ D477N in complex with Cg-Suv3$^{43-685}$) are deposited in the Protein Data Bank under accession numbers PDB: 6F3H and 6F4A. All other data are available from the corresponding author upon reasonable request.

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

## Acknowledgements

We thank Roman Szczęsny and Janusz Bujnicki for critically reading the manuscript, Prof. Dr Claudia Höbartner for helpful comments on alkaline hydrolysis ladders and Iwona Ptasiewicz for excellent technical assistance. This work was funded by National Science Center SYMFONIA grant (00463 to M.N. and P.G.), National Science Center FUGA grant (UMO-2012/04/S/NZ1/00036 to Z.W.) and National Science Center OPUS grant (2014/15/B/ST4/04839 to M.K.). This work was also supported by the Centre for Preclinical Research and Technology (European Union POIG.02.02.00-14-024/08-00). M.N. was a recipient of Ideas for Poland awards from the Foundation for Polish Science.

## Author contributions

M.R. prepared expression constructs, purified proteins, solved the crystal structures, designed and performed biochemical experiments with fluorescent substrates. Z.W. designed and performed biochemical experiments with radioactive substrates. M.T. executed SAXS measurements and analyzed the data. A.K., K.L.-D., A.K. performed yeast complementation experiments. J.P. prepared DNA constructs. M.C.-C. performed MALS experiments. E.N. participated in X-ray data collection and solution of the structures. P. G., M.K., A.D. and M.N. supervised the project. M.R., Z.W., A.D. and M.N. wrote the manuscript with input from all authors.

## Additional information

**Competing interests:** The authors declare no competing financial interests.

