## [Peer Review File · Nature Communications]

Reviewers' comments:

Reviewer #1 (Remarks to the Author):

The manuscript provides structural and mechanistic insight into the cooperative relationship between the mitochondrial proteins Dss1, a 3'-5' exonuclease, and Suv3, a ATP dependent RNA helicase. The authors present biochemical analysis showing that mtEXO activity requires ATP for substrate degradation. The channel which binds RNA is shown to harbour 13-17nt of ssRNA. More importantly, the crystal structure of mtEXO locates the points of contact between Suv3 and Dss1 to the RecA1 domain of Suv3 and the helix-turn-helix domain of Dss1. The structure is further analyzed by SAXS modelling and alignment with homologues.

Overall comments:

The science presented in this paper is sound, but the presentation might be better organised, and there are certain figures that would benefit from improvement. There are many expression constructs made, but descriptions are lacking for these. Many abbreviations are used as well, but some are unnecessary while some are vague.

Some of the interfacial mutations between the helicase and ribonuclease do not disrupt the complex but do impact on activity. This suggests that in addition to physical proximity of the helicase and ribonuclease activities, there must be some allosteric communication. This point seems important and might be noted more strongly in the discussion. Do the authors have a hypothesis for how this might work?

The identification of HTH and winged helix motifs in Dss1 is very interesting. It might be helpful to examine an electrostatics surface representation of the protein to see if there is a clear electropositive clustering that could aid in binding to nucleic acids.

Minor comments:

Summary, page 2

"The nuclease and helicase activities play pivotal roles in various aspects of RNA processing and degradation. These two activities are often assembled in multi-subunit complexes."

This would read better as

"Nuclease and helicase activities play pivotal roles in various aspects of RNA processing and degradation. These two activities are often present in different subunits of multi-subunit complexes from RNA metabolism."

Introduction

Page 3 line 38:

"The main executors of the RNA degradation are processive exoribonucleases that often form macromolecular assemblies and can act from either end of RNA molecules"

Change to

"The main executors of the RNA degradation in eukaryotes are processive exoribonucleases that often form macromolecular assemblies and can act from either end of RNA molecules"

Page 3 line 42: "help dissolve RNA secondary structures." Dissolve is inaccurately used. Perhaps use "unwind RNA".

Page 3 line 47:

"Both components are essential for the functioning of the complex in vivo, and"

Change to read

"The activities of both components are essential for the functioning of the complex in vivo, and"

Page 3 line 55: "...ATPase/helicase and nuclease activities." This sentence is slightly confusing. Is it referring to the general trend of ATPases associating with nucleases, or this specific case of Dss1/Suv3 association? Please be more specific.

Page 4 line 79: "RNA occupies..." Please specify RNA to be either double stranded or single stranded.

Page 4 line 80: "...active site located at its end..." Please specify which end in relation to the nucleic acid binding channel.

Page 5 line 87: "most likely evolved" This needs more verification. Perhaps removing it would eliminate any confusion.

Results

Page 5 line 92: "...Cg-Dss1" This is the first time the readers are introduced to the abbreviation Cg. It means the species *Candida glabrata*, but perhaps putting the abbreviation after the first appearance of the species name in page 5 line 85 would be helpful.

Page 5 line 96: Many N-terminal residues are deleted and the active site substitution is performed without explaining the reasoning. Please explain why those mutations were necessary or helpful. Also, a schematic of which domains these residues correspond to would also be helpful. Consider citing the schematic on figure 1.

Page 6 line 108: Please explain that a structural alignment was performed using DALI.

Page 6 line 114-122: This passage is unclear. There are a lot of comparisons to other WH domains, but no connection with Dss1 and explanation why the comparisons are important. This section mentions that WH domains are known to bind nucleic acids or mediate protein-protein interactions. Also, similar structures are found to bind DNA. Also, Cullin1 is a scaffold that binds the RING finger protein Rbx1. How are these structures important to Dss1? Perhaps leaving out this section would make the paragraph more clear. If necessary, include homologues only if they are important to the readers' understanding of the Dss1 WH domain.

Page 6 line 127-129: "Equivalent residues ... which could be involved in DNA binding." This sentence is unnecessary since the next sentence essentially states that the residues are not involved in DNA binding as seen on the structure. Including it may confuse the reader.

Page 7 line 148: Can a figure be made to highlight how the CSD1 has unique placement? Perhaps aligning it with other CSD1 homologues could be helpful. The statement that the Dss1 CSD1 has rotation of 90deg and translation of 15A would be visualized.

Page 8 line 157: Please show that the structures of Dss1 is similar to another member of the RNR family in an aligned figure.

Page 8 line 185-186: "Suv3 interacts with the funnel..." The funnel appears for the first time in this text at this position. Please explain where the funnel is on Dss1.

Page 9 line 206: Please cite figure 2C after "rotated by 90deg"

Page 11 line 243: "Furthermore, the N-terminal domain..." This is the same as ND as described earlier. Please keep nomenclature and abbreviations consistent to avoid confusion.

Page 11, line 247 "similar respiratory growth "might be more precise as "similar respiratory growth behaviour"

Page 12, line 256 "and showed that the Nterminal domain of Cg-Suv3 was mobile". would be better as "and are consistent with the N-terminal domain of Cg-Suv3 having mobility with respect to the core of the complex".

Page 12 line 269: Please describe the mtEXO channel in more detail. Include surrounding residues on surrounding domains if possible.

Page 13 line 282: "at later time-points" Please specify the time point.

Page 15-16, lines 337-355: These paragraphs might move into the discussion section.

Page 17 line 374: consider only using half-lives to quantify kinetics, as percentages can be confusing.

Figures

1. Consider making the sequence on top figure 1A. It may be easier to refer to it throughout the paper.
2. Please specify what diagram is meant by "diagram above" in the caption of figure 2A
3. Figure 4A is very helpful and allows the readers to follow the experiments.

Reviewer #2 (Remarks to the Author):

The manuscript "Structural analysis of mtEXO ... " by Razew and coworkers presents a detailed structure-function analysis of the mitochondrial degradosome. There are many interesting aspects that are unique to the mitochondrial complex, which makes this study interesting and complementary to other reports. Indeed, RNA degradation is an essential process that removes RNA molecules that could have toxic effects on the cell. Interestingly, the RNA degradation machineries seems to be invented several times, since they are present in all cells, prokaryotic or eukaryotic, in the different compartments in eukaryotic cells, and most importantly, they resemble each other in function, but not in the composition. The bacterial degradosomes are composed of RNases and DEAD-box RNA helicases, whereas the archaeal and eukaryotic exosomes are composed of RNases and Ski2-like helicases. In the work described here, the mitochondrial degradosome is composed of Dss1, a 3'-5' exoribonuclease, and Suv3, a RNA helicase conserved from fungi to humans, which is close to, but clearly distinct from Ski2-like RNA helicases.

The manuscript describes first the structure of Dss1, which to my knowledge is new. Although it has some classical domains, it is different from other described 3'-5' exoribonucleases. Most importantly it

carries two previously unidentified domains, where the second of them may be part of the interface with the Suv3 RNA helicase. The RNB domain is a classical RNase domain with a channel to accommodate a ssRNA.

The authors also present the structure of the mtEXO complex, composed of the RNase and the helicase (the title of this section: "Crystal structure of the mtEXO complex reveals the arrangement of complex subunits" is a little bit cumbersome and too "complex"). This complex reveals the interfaces of the two subunits. Although a mutational analysis does not disrupt the complex, these mutations cause respiratory deficiency in yeast. It is not clear, if these mutations reduce the affinity of the two subunits, which would no longer be sufficient *in vivo*, but still sufficient *in vitro* through the second interaction surface. Together with the structure of the human Suv3, the authors can clearly explain the 3'-5' exonuclease activity and the requirement for the helicase to provide a ssRNA.

To follow up of this crystallographic analysis, the authors then went on to confirm their data in solution by small-angle X-ray scattering.

The structure allows to estimate the length of the RNA within the mtEXO complex to be 18 nucleotides long. Using different approaches the authors confirm this length, also the experimental length is one nucleotide shorter, which is explained by the authors by breathing of the molecules.

Finally the authors produce biochemical experiments to show the enzymatic activities of the subunits either alone or in complex. Most importantly, the helicase activity depends on the presence of the complex and structured RNA can only be degraded by the mtEXO, and not by Dss1 alone.

Reviewer #3 (Remarks to the Author):

The authors report structural analyses of the yeast mitochondrial RNA degradation complex mtEXO and support their interpretation of the structure by *in vivo* mutational data, and biochemical analyses of RNA protection and RNase activity. The coupling of a helicase and RNase II family exonuclease fits a growing trend, but the mtEXO is significantly different from previously characterized structures

The structural analysis appears to be sound, the biochemistry is generally convincing and the results will be of wide interest.

Minor point:

1) Fig. 4d: This is the least convincing data presented. Looking at the gel, the RNase 1 protection patterns look very similar +/- Suv3, and do not correspond well with the predicted result in either case. The authors rationalize this outcome, but similar suggestions could have been made whatever the results had been. It is not clear that this panel is helpful. The use of the 5'-fluorescein-labeled substrate is a nice approach and appears to have given much clearer data.

Response to the reviewers

Structural analysis of mtEXO mitochondrial RNA degradosome reveals tight coupling of nuclease and helicase components

Razew M. et al.

We would like to thank the Reviewers for their constructive comments. We introduced changes to the presentation of our data which are described point-by-point below. All the changes in the manuscript text file are highlighted in blue.

Reviewer #1 (Remarks to the Author):

The manuscript provides structural and mechanistic insight into the cooperative relationship between the mitochondrial proteins Dss1, a 3'-5' exonuclease, and Suv3, a ATP dependent RNA helicase. The authors present biochemical analysis showing that mtEXO activity requires ATP for substrate degradation. The channel which binds RNA is shown to harbour 13-17nt of ssRNA. More importantly, the crystal structure of mtEXO locates the points of contact between Suv3 and Dss1 to the RecA1 domain of Suv3 and the helix-turn-helix domain of Dss1. The structure is further analyzed by SAXS modelling and alignment with homologues.

Overall comments:

The science presented in this paper is sound, but the presentation might be better organised, and there are certain figures that would benefit from improvement.

We are very happy to learn that this reviewer considers our science sound. Below we describe how, based on the comments of the reviewers, we improved the presentation of our data.

There are many expression constructs made, but descriptions are lacking for these. Many abbreviations are used as well, but some are unnecessary while some are vague.

We have added a new Supplementary Table 1 (page 12 of Supplementary Information) which lists all the expression constructs. We carefully verified that all the abbreviations are used in a consistent manner and we corrected their use where necessary.

Some of the interfacial mutations between the helicase and ribonuclease do not disrupt the complex but do impact on activity. This suggests that in addition to physical proximity of the helicase and ribonuclease activities, there must be some allosteric communication. This point seems important and might be noted more strongly in the discussion. Do the authors have a hypothesis for how this might work?

Yes, indeed, this is an important point. The fact that the two mutants affect the activity without disrupting the complex imply that the helicase and nuclease subunits must not only interact but also need to be properly aligned. This is a consequence of the uniquely tight structural and mechanistic coupling of the nuclease and helicase activities. This precise alignment is required for Suv3 to effectively feed the RNA into the channel in Dss1. In addition mtEXO complex needs to accommodate the conformational changes of the Suv3 helicase in the ATPase cycle (Fig. 6). If one of the contact

points between helicase and nuclease are lost and the rigidity of the complex is reduced, the movements of Suv3 can lead to even more pronounced misalignment of the two subunits and result in defects in mtEXO function. We added these considerations to the Discussion on page 16/17 (lines 353-379).

The identification of HTH and winged helix motifs in Dss1 is very interesting. It might be helpful to examine an electrostatics surface representation of the protein to see if there is a clear electropositive clustering that could aid in binding to nucleic acids.

This is a very helpful suggestion. We have added new panels to Figure 1 showing the electrostatic surface potential. This analysis did reveal a positively charged path on the surface of HTH domain. It is now described on page 7 (lines 137-140).

Minor comments:

Summary, page 2

"The nuclease and helicase activities play pivotal roles in various aspects of RNA processing and degradation. These two activities are often assembled in multi-subunit complexes."

This would read better as

"Nuclease and helicase activities play pivotal roles in various aspects of RNA processing and degradation. These two activities are often present in different subunits of multi-subunit complexes from RNA metabolism."

Corrected as suggested. We now state "nucleic acid metabolism" to underline the fact that this is a more universal phenomenon which also applies to DNA (page 2, lines 19-21).

Introduction

Page 3 line 38:

"The main executors of the RNA degradation are processive exoribonucleases that often form macromolecular assemblies and can act from either end of RNA molecules"

Change to

"The main executors of the RNA degradation in eukaryotes are processive exoribonucleases that often form macromolecular assemblies and can act from either end of RNA molecules"

Corrected as suggested.

Page 3 line 42: "help dissolve RNA secondary structures." Dissolve is inaccurately used. Perhaps use "unwind RNA".

Corrected as suggested.

Page 3 line 47:

"Both components are essential for the functioning of the complex in vivo, and"

Change to read

"The activities of both components are essential for the functioning of the complex in vivo, and"

Corrected as suggested (page 3, line 48).

Page 3 line 55: "...ATPase/helicase and nuclease activities." This sentence is slightly confusing. Is it referring to the general trend of ATPases associating with nucleases, or this specific case of Dss1/Suv3 association? Please be more specific.

Changed to "Biochemical studies revealed a remarkable functional interdependence of the nuclease and the helicase activities within mtEXO" (page 3, line 54).

Page 4 line 79: "RNA occupies..." Please specify RNA to be either double stranded or single stranded.

"Single-stranded RNA" added. The second part of the sentence now becomes redundant and was removed.

Page 4 line 80: "...active site located at its end..." Please specify which end in relation to the nucleic acid binding channel.

Together with the change from the previous point, the two sentences now read "Single-stranded RNA occupies the nucleic acid-binding channel inside the RNB domain. The active site is located at the opposite end of the channel relative to its entrance and interacts with the 3'-end of the substrate" (page 4, lines 79-81).

Page 5 line 87: "most likely evolved" This needs more verification. Perhaps removing it would eliminate any confusion.

Yes, indeed this is speculative. We changed the sentence to read "Our results revealed unique structural features of Dss1 that mediate a stable interaction with Suv3."

Results

Page 5 line 92: "...Cg-Dss1" This is the first time the readers are introduced to the abbreviation Cg. It means the species *Candida glabrata*, but perhaps putting the abbreviation after the first appearance of the species name in page 5 line 85 would be helpful.

Corrected as suggested (page 5, line 86).

Page 5 line 96: Many N-terminal residues are deleted and the active site substitution is performed without explaining the reasoning. Please explain why those mutations were necessary or helpful. Also, a schematic of which domains these residues correspond to would also be helpful. Consider citing the schematic on figure 1.

Yes, indeed, this is important information. We moved the relevant description from the Methods section to the results. Now the design of the crystallized construct is explained on page 5 (lines 96-102). We also marked the deleted N-terminal fragment of Dss1 and the positions of selected active site and RNA-binding residues in the diagram in Figure 1. The schematic of mtEXO constructs in Fig 2a was also changed to mark the fragments that were deleted.

Page 6 line 108: Please explain that a structural alignment was performed using DALI.

Corrected as suggested (page 6, line 114).

Page 6 line 114-122: This passage is unclear. There are a lot of comparisons to other WH domains, but no connection with Dss1 and explanation why the comparisons are important. This section mentions that WH domains are known to bind nucleic acids or mediate protein-protein interactions. Also, similar structures are found to bind DNA. Also, Cullin1 is a scaffold that binds the RING finger protein Rbx1. How are these structures important to Dss1? Perhaps leaving out this section would make the paragraph more clear. If necessary, include homologues only if they are important to the readers' understanding of the Dss1 WH domain.

We agree that this part of the text was not clear. We removed the description of the related structures which indeed did not provide additional new information about Dss1 WH domain and left only the information directly relevant to this domain (page 6/7, lines 122-131).

Page 6 line 127-129: "Equivalent residues ... which could be involved in DNA binding." This sentence is unnecessary since the next sentence essentially states that the residues are not involved in DNA binding as seen on the structure. Including it may confuse the reader.

The calculation of the electrostatic surface potential was a very helpful suggestion. It showed a presence of a positively charged patch in HTH domain which may participate in nucleic acid binding by Dss1 alone. We now show the electrostatic surface potential in Fig 1c and on page 7 (lines 137-140) we now state: "Indeed, this domain in Dss1 comprises a positively charged patch on its surface (made by residues Lys387 and Arg390) that could aid in binding of nucleic acids (Fig. 1c). However, in the mtEXO complex, the residues forming this patch participate in protein-protein interactions (Fig. 2a-c) and are unlikely to interact with the nucleic acid."

Page 7 line 148: Can a figure be made to highlight how the CSD1 has unique placement? Perhaps aligning it with other CSD1 homologues could be helpful. The statement that the Dss1 CSD1 has rotation of 90deg and translation of 15A would be visualized.

Prompted by this comment we performed additional analyses of the N-terminal domain of Dss1. We found that although the region of Dss1 corresponding to CSD1 in other RNB enzymes in Dss1 adopts a similar β -barrel fold (with lack of significant sequence homology which is often observed for the Cold Shock Domains), it resembles more closely KOW domains. The KOW sequence motif is absent, however, so we chose to name the first N-terminal domain of *Cg*-Dss1 " β -barrel". Interestingly similar β -barrel domain is present in the exosome associated Ski2 helicase. We added the description of this analysis to the text on page 6 (lines 111-119). In Supplementary Fig. 2 we now use different colors to distinguish CSD1 and β -barrel domains. Given the relatively distant relationship between CSD1 and the β -barrel domain we would prefer not to prepare a detailed analysis of the difference in their placement in various structures. This change in domain terminology does not affect any of our conclusions.

Page 8 line 157: Please show that the structures of Dss1 is similar to another member of the RNR family in an aligned figure.

We changed the Supplementary Figure showing describing the RNA binding by Dss1 (Supplementary Fig. 4). It now contains in panel (b) a superposition of RNase II with selected residues involved in RNA binding and cleavage shown. Furthermore, Supplementary Fig. 3 shows a superposition and a structure-based sequence alignment of the conserved RNB domain of the RNR family proteins in which we indicate the active site and RNA-binding residues.

Page 8 line 185-186: “Suv3 interacts with the funnel...” The funnel appears for the first time in this text at this position. Please explain where the funnel is on Dss1.

The sentence was rewritten to: “In the *Cg*-mtEXO structure, Suv3 is positioned close to the small accessory domains of Dss1: β -barrel, HTH, WH, and S1 which decorate the catalytic RNB domain and form a funnel around the entry to the RNA-binding channel” (page 9, lines 196-198).

Page 9 line 206: Please cite figure 2C after “rotated by 90deg”

Corrected as suggested. After adding an additional panel in Figure 2 now we cite Figure 2d (page 10, line 219).

Page 11 line 243: “Furthermore, the N-terminal domain...” This is the same as ND as described earlier. Please keep nomenclature and abbreviations consistent to avoid confusion.

We now consistently use “ND” thorough the text.

Page 11, line 247 “similar respiratory growth ”might be more precise as “similar respiratory growth behaviour”

Corrected as suggested (page 12, line 255).

Page 12, line 256 “and showed that the Nterminal domain of *Cg*-Suv3 was mobile”. would be better as “and are consistent with the N-terminal domain of *Cg*-Suv3 having mobility with respect to the core of the complex”.

Corrected as suggested.

Page 12 line 269: Please describe the mtEXO channel in more detail. Include surrounding residues on surrounding domains if possible.

We now refer to the description of RNA-binding residues for Dss1 on page 12/13 (lines 272-274) “A 6 nt RNA fragment is present in the *Cg*-mtEXO structure inside the RNA-binding channel of Dss1 and it is superimposable with the RNA observed in *Cg*-Dss1 forming the same contacts with the protein (Supplementary Fig. 4b, c, Supplementary Note 1).” For *Hs*-Suv3 we now state (page 13, lines 276-279) “The RNA is bound by residues from the helicase motifs: Ia (Lys234), Ib (Thr275) of the RecA1 domain and IV (Phe373 and Lys375), V (Thr424 and Asp425) of the RecA2 domain⁷. These motifs are conserved in *Cg*-Suv3 and we assume they will contact RNA in the same way.”

Page 13 line 282: “at later time-points” Please specify the time point.

Corrected as suggested (page 13, line 294).

Page 15-16, lines 337-355: These paragraphs might move into the discussion section.

These paragraphs indeed fit better in the discussion and they were moved there (page 16, lines 353-369). The panels showing the proposed movements of Suv3 within mtEXO are now shown in a separate Fig. 6.

Page 17 line 374: consider only using half-lives to quantify kinetics, as percentages can be confusing.

Corrected as suggested.

Figures

1. Consider making the sequence on top figure 1A. It may be easier to refer to it throughout the paper.

The sequence of Dss1 is rather long and could not be legible in Fig 1. Instead we now show a sequence alignment of RNase II family nucleases RNB domain in Supplementary Fig. 3. We also marked the positions of the residues forming the active site and selected amino acids involved in RNA binding in the diagram in Fig 1.

2. Please specify what diagram is meant by “diagram above” in the caption of figure 2A

This part of the legends was rewritten to “Schematic of the domain composition of *Cg*-Dss1 and *Cg*-Suv3. The dashed line corresponds to the regions deleted in the crystallized variants.”

3. Figure 4A is very helpful and allows the readers to follow the experiments.

Yes, indeed this is a complex experiment. Thank you.

Reviewer #2 (Remarks to the Author):

The manuscript “Structural analysis of mtEXO ... “ by Razew and coworkers presents a detailed structure-function analysis of the mitochondrial degradosome. There are many interesting aspects that are unique to the mitochondrial complex, which makes this study interesting and complementary to other reports. Indeed, RNA degradation is an essential process that removes RNA molecules that could have toxic effects on the cell. Interestingly, the RNA degradation machineries seems to be invented several times, since they are present in all cells, prokaryotic or eukaryotic, in the different compartments in eukaryotic cells, and most importantly, they resemble each other in function, but not in the composition. The bacterial degradosomes are composed of RNases and DEAD-box RNA helicases, whereas the archaeal and eukaryotic exosomes are composed of RNases and Ski2-like helicases. In the work described here, the mitochondrial degradosome is composed of Dss1, a 3'-5' exoribonuclease, and Suv3, a RNA helicase conserved from fungi to humans, which is close to, but clearly distinct from Ski2-like RNA helicases.

The manuscript describes first the structure of Dss1, which to my knowledge is new. Although it has some classical domains, it is different from other described 3'-5' exoribonucleases. Most importantly it carries two previously unidentified domains, where the second of them may be part of the

interface with the Suv3 RNA helicase. The RNB domain is a classical RNase domain with a channel to accommodate a ssRNA.

The authors also present the structure of the mtEXO complex, composed of the RNase and the helicase (the title of this section: “Crystal structure of the mtEXO complex reveals the arrangement of complex subunits” is a little bit cumbersome and too “complex”).

The title of this section was changed to “Structure of mtEXO reveals the arrangement of the subunits” (page 8, line 174).

This complex reveals the interfaces of the two subunits. Although a mutational analysis does not disrupt the complex, these mutations cause respiratory deficiency in yeast. It is not clear, if these mutations reduce the affinity of the two subunits, which would no longer be sufficient in vivo, but still sufficient in vitro through the second interaction surface. Together with the structure of the human Suv3, the authors can clearly explain the 3'-5' exonuclease activity and the requirement for the helicase to provide a ssRNA.

To follow up of this crystallographic analysis, the authors then went on to confirm their data in solution by small-angle X-ray scattering.

The structure allows to estimate the length of the RNA within the mtEXO complex to be 18 nucleotides long. Using different approaches the authors confirm this length, also the experimentally length is one nucleotide shorter, which is explained by the authors by breathing of the molecules.

Finally the authors produce biochemical experiments to show the enzymatic activities of the subunits either alone or in complex. Most importantly, the helicase activity depends on the presence of the complex and structured RNA can only be degraded by the mtEXO, and not by Dss1 alone.

Reviewer #3 (Remarks to the Author):

The authors report structural analyses of the yeast mitochondrial RNA degradation complex mtEXO and support their interpretation of the structure by in vivo mutational data, and biochemical analyses of RNA protection and RNase activity. The coupling of a helicase an RNase II family exonuclease fits a growing trend, but the mtEXO is significantly different from previously characterized structures

The structural analysis appears to be sound, the biochemistry is generally convincing and the results will be of wide interest.

We are very glad to hear that this referee considers our data solid and results interesting.

Minor point:

1) Fig. 4d: This is the least convincing data presented. Looking at the gel, the RNase 1 protection patterns look very sim +/- Suv3, and do not correspond well with the predicted result in either case. The authors rationalize this outcome, but similar suggestions could have been made whatever the results had been. It is not clear that this panel is helpful. The use of the 5'-fluorescein-labeled substrate is a nice approach and appears to have given much clearer data.

Indeed, the results of the footprinting are more difficult to interpret than those obtained with 5'-fluorescein-labeled substrate. Following this comment we decided to remove the footprinting result from the main Figure 4. We would prefer, however, to leave the second footprinting result in the Supplementary Information. In our view it provides valuable information. Firstly, the footprint for mtEXO is strictly ATP-dependent which proves that the RNA protection is specific and lends further support for the active RNA feeding mechanism we propose. Moreover, the surface potential calculation for Dss1 revealed a positively charged patch in HTH domain (Fig. 1c). This patch is likely to bind the RNA which would explain larger than expected footprint for Dss1 alone, which lends further support to our interpretation. We rearranged the text and included a short footprint description in the section "mtEXO but not Dss1 alone degrades structured RNAs in an ATP-dependent manner" (page 14, line 304). A more detailed description and interpretation of the footprinting data is now presented in the Supplementary Note 4.

Additional changes:

Supplementary Table 1 "Data collection and refinement statistics" was moved to the main text.

Reference 24 was changed to Nguyen, Y. et al. Structural and mechanistic roles of novel chemical ligands on the SdiA quorum-sensing transcription regulator. *MBio* 6(2015).

On page 7 (line 140) term "of unknown function" was removed since the HTH domain in DrII forms an open RNA-binding surface.